# Metabolomic Plasma Profile of Chronic Obstructive Pulmonary Disease Patients

**DOI:** 10.3390/ijms26104526

**Published:** 2025-05-09

**Authors:** Carme Casadevall, Bella Agranovich, Cesar Jesse Enríquez-Rodríguez, Rosa Faner, Sergi Pascual-Guàrdia, Ady Castro-Acosta, Ramon Camps-Ubach, Judith Garcia-Aymerich, Esther Barreiro, Eduard Monsó, Luis Seijo, Juan José Soler-Cataluña, Salud Santos, Germán Peces-Barba, José Luis López-Campos, Ciro Casanova, Alvar Agustí, Borja G. Cosío, Ifat Abramovich, Joaquim Gea

**Affiliations:** 1Hospital del Mar Research Institute, Servei de Pneumologia, Hospital del Mar, MELIS Department, Universitat Pompeu Fabra, 08013 Barcelona, Spain; carme.casadevall@upf.edu (C.C.); cesarjesse.enriquez01@alumni.upf.edu (C.J.E.-R.); ramon.camps.ubach@hmar.cat (R.C.-U.);; 2Centro de Investigación Biomédica en Red, Área de Enfermedades Respiratorias (CIBERES), Instituto de Investigación Carlos III (ISCiii), 28029 Madrid, Spain; rfaner@recerca.clinic.cat (R.F.); lseijo@unav.es (L.S.); juanjosoler1965@gmail.com (J.J.S.-C.); saludsantos@bellvitgehospital.cat (S.S.); gpecesbarba@gmail.com (G.P.-B.); lcampos@separ.es (J.L.L.-C.); casanovaciro@gmail.com (C.C.); aagusti@clinic.cat (A.A.); borja.cosio@ssib.es (B.G.C.); 3The Ruth and Bruce Rappaport Faculty of Medicine, Technion, Israel Institute of Technology, Haifa 3525433, Israel; bellakr@technion.ac.il (B.A.); ifat.a@technion.ac.il (I.A.); 4Departament de Biomedicina, Universitat de Barcelona, 08007 Barcelona, Spain; 5Fundació Clínic per la Recerca Biomèdica (FCRB), IDIBAPS, 08036 Barcelona, Spain; 6Servicio de Neumología, Hospital 12 de Octubre, 28041 Madrid, Spain; ady@h12o.es; 7MELIS Department, Universitat Pompeu Fabra, 08003 Barcelona, Spain; judith.garcia@isglobal.org; 8ISGlobal, 08003 Barcelona, Spain; 9Centro de Investigación Biomédica en Red, Área de Epidemiología y Salud Pública (CIBERESP), ISCiii, 28029 Madrid, Spain; 10Fundació Institut d’Investigació i Innovació Parc Taulí (I3PT), 08208 Sabadell, Spain; 11Servicio de Neumología, Clínica Universidad de Navarra, 28027 Madrid, Spain; 12Servicio de Neumología, Fundación Jiménez Díaz, Universidad Autónoma de Madrid, 28049 Madrid, Spain; 13Servicio de Neumología, Hospital Arnau de Vilanova-Lliria, Universitat de València, 46015 Valencia, Spain; 14Servei de Pneumologia, Fundació Institut d’Investigació Biomèdica de Bellvitge (IDIBELL), Universitat de Barcelona, 08908 Hospitalet, Spain; 15Unidad Médico-Quirúrgica de Enfermedades Respiratorias, Hospital Universitario Virgen del Rocío, Universidad de Sevilla, 41012 Sevilla, Spain; 16Servicio de Neumología-Unidad de Investigación Hospital Universitario La Candelaria, Universidad de La Laguna, 38010 Tenerife, Spain; 17Servei de Pneumologia (Institut Clínic de Respiratori), Hospital Clínic, 08036 Barcelona, Spain; 18Servicio de Neumología, Hospital Son Espases, Institut d’Investigació Sanitària Illes Balears (IdISBa), Universitat de les Illes Balears, 07120 Palma, Spain

**Keywords:** COPD, metabolites, lipid homeostasis, amino acids, fatty acids, acylcarnitines

## Abstract

The analysis of blood metabolites may help identify individuals at risk of having COPD and offer insights into its underlying pathophysiology. This study aimed to identify COPD-related metabolic alterations and generate a biological signature potentially useful for screening purposes. Plasma metabolomic profiles from 91 COPD patients and 91 controls were obtained using complementary semi-targeted and untargeted LC-MS approaches. Univariate analysis identified metabolites with significant differences between groups, and enrichment analysis highlighted the most affected metabolic pathways. Multivariate analysis, including ROC curve assessment and machine learning algorithms, was applied to assess the discriminatory capacity of selected metabolites. After adjustment for major potential confounders, 56 metabolites showed significant differences between COPD patients and controls. The enrichment analysis revealed that COPD-associated metabolic alterations primarily involved lipid metabolism (especially fatty acids and acylcarnitines), followed by amino acid pathways and xenobiotics. A panel of 10 metabolites, mostly related to lipid metabolism, demonstrated high discriminatory performance for COPD (ROC-AUC: 0.916; 90.1% sensitivity and 89% specificity). These findings may contribute to improving screening strategies and a better understanding of COPD-related metabolic changes. However, our findings remain exploratory and should be interpreted with caution, needing further validation and mechanistic studies.

## 1. Introduction

Chronic obstructive pulmonary disease (COPD) is a very prevalent disorder, mainly characterized by respiratory symptoms, chronic poorly reversible and often progressive airflow obstruction [1] as well as heterogeneous clinical presentations. COPD is the result of long-term exposure to harmful inhaled noxious agents, such as tobacco smoke or ambient particulates, and is associated with an abnormal pulmonary and systemic inflammatory response [2,3]. However, and despite its high prevalence, COPD also maintains a high rate of underdiagnosis [4,5,6,7,8,9,10,11,12], which is probably due to a low clinical suspicion and the need for a technically well-performed forced spirometry. The latter is a well-established technique, but its correct use requires a minimal infrastructure, and skills and technical requirements that are difficult to be extended to primary medicine and/or very large populations. Therefore, any approach that facilitates the suspicion of COPD, focusing the performance of spirometry on high-risk populations, could facilitate diagnosis.

Growing evidence suggests that local and systemic inflammation are key contributing factors to COPD progression and its comorbidities [1,3,13]. The local inflammatory response to environmental cues, triggered to eliminate harmful stimuli and promote lung tissue repair and remodeling, also contributes to altering the dynamic equilibrium of the respiratory microbiome composition (dysbiosis). Microbiome dysbiosis further affects the immune system balance enhancing lung and peripheral inflammatory responses [14,15,16]. In turn, the modulation of the immune system is underlined by metabolic adaptation in local and systemic immune cells [17]. Thus, a deeper understanding of the metabolic rearrangement occurring in COPD is being provided by recent research [17,18,19,20,21].

Metabolomics is a useful scientific tool that helps to identify metabolite profiles and enables our understanding of the pathogenic mechanisms of complex diseases such as COPD. Even though a growing body of evidence highlights the metabolic abnormalities in this disorder [17,18,19,20,21], most of the previous studies are limited by the small sample size, a preponderance of elderly patients or fail to consider potential confounding variables, such as smoking, nutritional status and the multimorbidity frequently present in these patients. Moreover, most of these studies have used hypothesis-driven approaches directed at specific candidate metabolites. In the present study, we employed semi-targeted and untargeted complementary approaches coupled with artificial intelligence methodology. The aim was to compare the plasma metabolomes of a wide range of COPD patients and matched controls (HC, asymptomatic smokers without airflow limitation) to identify differential metabolite concentrations and their corresponding pathways, considering the main potential confounding factors. Therefore, the main objectives of this study were to gain valuable insight into the pathophysiology of the disease and to obtain a COPD metabolic risk signature, which may have a potential use for screening purposes.

## 2. Results

### 2.1. Main Characteristics of Participants

The general and clinical characteristics of the participants are summarized in Table 1. Briefly, both COPD and HC were medium-aged individuals, with similar sex distribution (minor male predominance), and mildly overweight. By study design, COPD patients presented an abnormal respiratory function with airway obstruction and reduced carbon monoxide diffusing capacity (DLco), with most of them classified as GOLD 2, followed by GOLD 3 and GOLD 1.

### 2.2. Metabolite Profile in COPD Patients

The liquid chromatography/mass spectrometry (LC-MS)/MS platform used in this study measured a total of 461 compounds of already known identity but, and as discussed in the Methods Section, only those metabolites identified in at least 80% of plasma samples were included in the final analysis, which encompassed 360 metabolites (78%).

From these 360 metabolites, the univariate analysis identified 74 that differed significantly (differentially abundant metabolites or DAMs) between HC and COPD patients, and of these metabolites, 54 and 20 were found to be over and underrepresented, respectively, in COPD patients (Figure 1). The highest proportion of DAMs was found in lipids and lipid-like molecules (29, 39.2%), followed by xenobiotics (13, 17.6%), amino acids and their analogues and derivatives (10, 13.5%) and carbohydrates and their conjugates (6, 8.1%). The detailed information on these metabolites is summarized in the Appendix A.

After data adjustment for potential confounding factors [age, gender, body mass index (BMI) and the intensity of recent smoke exposure], the concentration of 56 metabolites remained significantly different between patients and HC, 51 of which (91%) coincided with those from the former non-adjusted analysis (Figure 2; Appendix A). Based on the adjusted analysis, COPD patients were characterized by the following: (1) increased abundance of nine short- and medium-chain fatty acids (SCFAs and MCFAs, respectively) and derivatives. SCFAs included butyric, valeric and β-hydroxyisovaleric acids, whereas MCFA were 2-hydroxyisocaproic (HICA, metabolite of leucine), 3-methylglutaconic, caprylic, capric, caproleic and 5-dodecanoic acids; (2) higher levels of five acylcarnitines: acetyl-L-Carnitine, octanoylcarnitine, decanoylcarnitine, lauroylcarnitine and palmitoylcarnitine; and, (3) decreased concentration of long- and very long-chain fatty acids (LCFAs and VLCFAs, respectively): 2-hydroxymyristic, pentadecanoic, palmitic and 14-methylhexadecanoic, nonadecanoic, arachidic, behenic and lignoceric. By contrast, the DAMs more sensitive to the adjustment for potential confounding factors were amino acids (as well their analogues and derivatives) and xenobiotics.

Role of smoking status. Metabolite shifts were also compared between current and former smokers within each group. An elevated level of cotinine was the only marker discriminating smoking status in HC, whereas theophylline, didemethy-lisoproturon and 3-carboxy-4-methyl-5-propyl-2-furanpropionate (CMPF) differentiate current and former smokers.

### 2.3. Pathway Enrichment Analysis of DAMs in COPD

In order to explore the metabolic pathway perturbations associated with the disease, the 56 DAMs that varied significantly between the two groups after adjustments were mapped through Kyoto Encyclopedia of Genes and Genomes (KEGG) and Human Metabolome Database (HMDB) databases in 17 and 21 metabolic pathways, respectively. We found that lipid and amino acid metabolism emerged as the most significantly altered pathways in COPD (Appendix A).

### 2.4. Multivariate Analysis and AI Modeling

We again considered only the 56 metabolites that distinguished patients from HC after adjustments to evaluate key metabolites influencing COPD discrimination as potential biomarkers. A predictive model was established using the artificial intelligence (AI) Support Vector Machine (SVM) algorithms and Receiver Operating Characteristic (ROC) curves to explore which metabolite signature has the best sensitivity and specificity balance (Figure 3a). Among those biomarkers most strongly associated with COPD, SVM selected 10 metabolites for the scoring system (Table 2), which achieved an area under the curve (AUC) of 0.927 (Figure 3b). Notably, four of them (1-tetradecylamine, 2-naphthalenesulfonic, 4-dodecylbenzenesulfonic acids and pentapropylen glycol (PPG n5)) were considered as not being produced naturally by human beings (xenobiotics), but derived mainly from body care or cleaning products or potential products of microbiota.

### 2.5. Exclusion of Xenobiotics

Accordingly, after excluding these xenobiotics, we repeated the SVM analysis to identify a truly endogenous metabolic signature contributing to COPD discrimination but with clearer human biological origin. This new model identified 10 DAMs strongly associated with COPD (Table 3, where four new metabolites have substituted the former four xenobiotics), including palmitic, 14-methylhexadecanoic (derived from the previous one, where a methyl group has entered in the molecule), 2-hydroxytetradecanoic, glyceric and 2-aminonicotinic acids, as well as urocanate were significantly downregulated in COPD patients, whereas the remaining differential metabolites, gluconic and HICA acids, diethanolamine and N-methylglutamate) were upregulated (Table 3).

These biomarkers also showed a high discriminative value, with an AUC of 0.916 (Figure 4a,b).

## 3. Discussion

The main observations of the present study suggest the following: (1) metabolomics can reliably help to differentiate COPD patients from controls; (2) DAMs were mostly related to lipid metabolism, followed by that of proteins, and to a much lesser extent to carbohydrates and their conjugates, and tricarboxylic or citric acid cycle (TCA) intermediates; and, finally, (3) a group of ten metabolites may conform a predictive model, which is highly sensitive and specific for COPD. These findings will require further validation, ideally in independent cohorts and through longitudinal or mechanistic studies.

### 3.1. Previous Studies

Some previous studies have reported metabolic abnormalities in COPD [17,18,19,20,21,22]. However, they generally included a small number of subjects, with a lack of full clinical details and did not consider potential confounding factors that are frequently present in these patients and can influence metabolism. Moreover, they used hypothesis-driven approaches directed to specific candidate metabolites. Our study addresses these potential limitations by studying a larger population of well characterized COPD patients and HC, considering potential confounding factors in the analysis, and using a double and complementary not-hypothesis-driven metabolomic approach (semi-targeted plus untargeted assessments) coupled with supervised machine learning (IA) analysis to characterize and model the circulating metabolite changes more characteristic of COPD patients.

### 3.2. Interpretation of Novel Findings

#### 3.2.1. Lipids and Lipid-like Molecules

Lipids have three main functions: membrane building, energy storage and production, and signal transduction [23]. Moreover, they can be involved in either anabolic or catabolic pathways, such as those involved in the synthesis of new and more complex molecules and those derived from their oxidation (which in turn, can generate new metabolites and/or produce more energy). In the present study, close to half the DAMs were lipids, and more specifically fatty acids, their conjugates and esters. Fifteen were more abundant in COPD than in controls [including products of the leucine degradation process (HICA, β-hydroxyvaleric acid and 3-methylglutaconate), participants in the synthesis of other fatty acids (butyric, octanoic and decanoic acids), possible products of gut microbiota (pentanoate) and fatty acid esters (different acylcarnitines)], whereas ten more had lower levels in patients when compared to HC. While most of the under-represented fatty acids (such as lignoceric, behenic and docosatetraenoic) are involved in the biosynthesis of unsaturated fatty acids, others also participate in the synthesis of steroids, as is the case of palmitic acid, or eicosanoids, as for the arachidic acid. Moreover, four of them were integrated into the final COPD signature (palmitic, leucic/HICA, 2-hydroxymyristic and 14-methylhexadecanoic acids). Taken together, these findings may suggest that lipid metabolism is altered in COPD and could play a role in covering the increased energetic demands of these patients, as well as in modulating proinflammatory pathways through changes in eicosanoids (i.e., prostaglandins, thromboxanes and leukotrienes) production [24,25]. These lipid mediators are potent biological signaling molecules that can promote inflammation and hinder tissue repair [26]. The increased levels of diethanolamine (DEA, present in the COPD signature), involved in the glycerophospholipid metabolism, might be related to long-term effects of previous exacerbations [27]. Other authors have also highlighted the relevance of lipids in COPD pathophysiology [28,29]. Overall, our results are consistent with the hypothesis that lipid metabolic pathways are altered in COPD and that such alterations could affect cellular physiology, including immune and inflammatory responses, amongst others, thereby potentially contributing to disease development and persistence [24]. Nonetheless, these hypotheses warrant confirmation in future mechanistic and longitudinal studies.

#### 3.2.2. Xenobiotics

Of particular interest is the presence in plasma of several metabolites that are not endogenously produced by humans. Moreover, some of these xenobiotics were differentially abundant in COPD patients and HC. The interpretation of these findings remains challenging due to the limited information available on the biological roles of many xenobiotics, which often originate from diverse environmental, dietary or microbiota-associated sources. While some of these compounds could potentially reflect environmental exposures or microbial metabolism, their precise origin and role in COPD pathophysiology remains uncertain. In our dataset, eight xenobiotics were found in higher concentrations in COPD patients. These included compounds commonly used in personal care products or food additives [e.g., PPG n5, tetrapropylene glycol (PPG n4), 1-tetradecylamine and methyl vanillate], as well as industrial agents [e.g., bis(methylbenzylidene)sorbitol and tetraglyme], the latter of which has been associated with oxidative and genotoxic effects in vitro [30]. Additionally, we detected four xenobiotics considered to be environmental contaminants, primarily plasticizers or detergent components [e.g., 4-dodecylbenzenesulfonic acid, 2-naphthalenesulfonic acid, diisopropylethylamine (DIPEA) and bis(2-ethylhexyl) phthalate (DEHP)]. Although none of these compounds are known drug metabolites, the possibility of their presence as excipients, contaminants from inhalation devices, or components of pharmaceutical capsules used by patients cannot be fully excluded. Interestingly, three xenobiotics elevated in COPD samples may be of microbial origin, consistent with growing evidence on microbiota–host interactions in systemic inflammation [14,31]. For example, NP-013736, a surfactant-like compound, may be produced by *Stenotrophomonas rhizophila* [32], a Gram-negative bacterium found in food sources and potentially in the human gut. Similarly, 4-hydroxybenzaldehyde (4HBA) is known to be synthesized by various bacterial species [33], and 3-hydroxybenzaldehyde (3HBA), although produced by gut epithelial cells, may serve as a substrate or signaling molecule for microbial metabolism [34,35].

Despite these intriguing findings, we have chosen to exclude xenobiotics from the core mechanistic analysis of disease-related pathways, as our primary objective is to focus on endogenously regulated metabolites, which are more directly linked to intrinsic physiological and pathophysiological processes in COPD. Nevertheless, the altered abundance of several xenobiotics in COPD patients warrants further investigation. Future studies should aim to validate these associations and clarify their potential role, either as passive exposure markers or active modulators in disease development and/or progression.

#### 3.2.3. Amino Acids, Analogues and Derivates

The third most important metabolite class, although representing a much smaller number of DAMs (just 10.7% of the total) than the two preceding groups, was that of amino acids and associated molecules. Amino acids are the structural base of proteins, but they also participate in many biological functions such as the synthesis of serotonin, active amines, porphyrins, nitrogenous bases (and therefore, nucleic acids) and even nitric oxide. Four of these amino acids and related molecules, including tyrosine and one product of its metabolism (4-coumarate), as well as two metabolites of glutamine and branched-chain amino acids (N-methylglutamate and 3-hydroxybutanoate, respectively) were overrepresented in COPD patients. Only two others, associated with the metabolism of leucine and histidine (N-acetylleucine and urocanate, respectively), were under-expressed in patients. Moreover, two of these metabolites, N-methylglutamate and urocanate, were also present in the final COPD signature.

Overall, these findings may reflect the well-known predominance of increased proteolysis in COPD patients [36,37,38,39]. This process, which is thought to exceed protein synthesis in many cases, has been associated with systemic manifestations of the disease (such as low body weight and muscle dysfunction) [40], and may be partially driven by the need to generate energy to meet the increased metabolic requirements observed in COPD, both at rest and during exercise [41]. This hypermetabolic state is particularly pronounced in malnourished patients [42]. Additionally, lung and airway tissue remodeling during disease progression may represent a complementary source of protein degradation metabolites in COPD patients [38]. The ubiquitin–proteasome system appears to play a major role in the increased protein turnover [36], although other mechanisms, such as autophagy and oxidative stress, may also contribute to the presence of products of protein degradation products in the blood of such patients [39].

Another noteworthy observation of the present study is the increase in several carnitine-derived metabolites. Carnitines play a critical role in the transport of LCFA into mitochondria to be used in β-oxidation. Most of the carnitine-derived metabolites that were more abundant in the blood of our patients than in controls were linked to MCFAs or LCFAs metabolism. However, as previously mentioned, LCFAs and VLFAs were decreased in COPD patients. Taken together, these results may be consistent with the hypothesis of reduced energy production in COPD. Nonetheless, further mechanistic studies will be required to confirm these potential links and better understand their contribution to disease pathophysiology. Finally, the present results are in line with our findings in three recent studies, which showed that different combinations of proteins and their fractions, particularly those related to immune and hemostasis pathways, could serve as useful markers of either acute exacerbations, the frequent exacerbator phenotype and even poor vital prognosis in COPD [43,44,45].

#### 3.2.4. Carbohydrates

Sugars are very relevant molecules that can be catabolized to rapidly obtain energy (glycolysis), but are also components of coenzymes [such as adenosine triphosphate (ATP) and nicotinamide adenine dinucleotide (NAD)], nucleic acids, power storages (such as glycogen), and also include mucopolysaccharides of the connective tissue. Moreover, carbohydrates can also be associated with proteins and lipids (glycoproteins and glycolipids, respectively), with which they share their participation in the TCA and the mitochondrial respiratory chain. In fact, carbohydrates play relevant roles not only in the production and storage of energy, but also in the immune and inflammatory responses, blood clotting and lubrification of joints, among other relevant biological processes. Even though abnormalities in carbohydrate metabolism in COPD seem to be less prominent than those observed in lipid and protein pathways, some abnormalities have been previously described. For example, glycoprotein Glyc A, a marker of low-grade inflammation, is increased in COPD patients [46], although some N-acetyl glycoproteins are decreased in their blood. Furthermore, certain sugars (such as fructose, ribose, fucose) or their associated metabolites (i.e., glucosaminic acid) seem to be increased in patients with poor vital prognosis or in those with chronic bronchitis coexisting with lung cancer (turanose) [44,47,48,49]. Carbohydrate metabolism also appears to be particularly altered in more severely affected patients [50,51], and during exacerbations [52,53]. Moreover, increased levels of sialic acid in sputum have been reported as markers of either acute exacerbations or the frequent exacerbator phenotype [54]. In a previous study, we demonstrated that the efficiency of the lectin pathway of the complement system, which relies in carbohydrate recognition on bacterial surfaces, appears to be impaired in COPD patients experiencing an acute exacerbation [45]. Finally, some other carbohydrates (glucose, manose) have been shown to be decreased in the blood of patients with the eosinophilic phenotype [51].

Two of the metabolites chosen for our COPD signature are associated with the pentose phosphate pathway, which is closely linked to glycolysis and plays a key role in the synthesis of nucleotides and nucleic acids, as well as in the anabolic actions of nicotinamide adenine dinucleotide phosphate (NADPH). This pathway may also contribute to the synthesis of fatty acids, and hence to the formation of more complex lipids and to energy storage/production. Therefore, the altered levels of gluconic and glyceric acids observed in our study could point toward dysregulations in these interconnected biological processes [51]. However, these interpretations should be considered as hypothesis, and further studies are needed to confirm their functional implications in the context of COPD.

### 3.3. Potential Confounding Factors

We adjusted our results by some of the factors that are already known to intrinsically influence metabolism; gender, age, nutritional status and tobacco exposure [47].

In an interesting report from Liu et al., nutritional status was the most important of these cofactors in adult smokers, showing a close relationship with some infectious/inflammatory markers such as white cell count, C-reactive protein (CRP) and fibrinogen [55]. Gender also appears to have a strong influence on COPD metabolism [56] with females showing a higher metabolic dysregulation than males, and specifically in those metabolites that participate in the redox balance [57] and some cytokines [58]. Liu et al. showed an association of the female gender with 11-dehydrothromboxane B2 (11-dehTxB2), considered as an index of synthesis of thromboxane A2, which is involved in platelet aggregation [55]. Moreover, other markers seem to be lower in both healthy women and women with COPD, which is the case of the transcription factor peroxisome proliferator-activated receptor gamma (PPAR-γ) that regulates the immune responses and may be related to drop in estrogen levels in elderly women [59]. Interestingly, gender influences the metabolic response to exercise in COPD, and more specifically in metabolites linked to fatty acids (acetyl-CoA, oleic acid), proteins (glutamine, tryptophan, branched chain amino acids) and the TCA cycle (succinate, creatinine and α-ketoglutarate or a-KG) [60]. Aging is another physiological circumstance that induces relevant and specific changes in the body metabolism, with some of these changes appearing to differ between healthy individuals and COPD patients. For instance, this is the case of some amino acids and lipids that seem to be overproduced in elderly patients when compared with age-matched controls. Moreover, some of the changes linked to aging seem to appear earlier in COPD patients than in healthy individuals [22].

Finally, various studies have demonstrated that the smoking habit per se can modify the metabolism, since it is associated with inflammation, oxidative stress, platelet activation and increases in both lipid oxidation and mitochondrial respiration [61,62,63,64,65,66]. Many of the metabolites involved in these processes become modified by tobacco smoking. This is the case of increases in the von Willebrand factor (vWF, an endothelial factor that participates in the initial steps of hemostasis and is also a marker of low-degree inflammation) [55], diverse branched-chain and others amino acids [55,67,68], as well as eicosanoids [55,67,69], fibrinogen [70] or even some ions [71] shown by active smokers. Other abnormalities have also been related to passive smoking, or recent alternatives to classical tobacco smoking. The former included alterations in phospholipid, amino acid/peptide and purine metabolism [72,73], whereas methylation of various molecules have also been recently reported in users of the latter [74]. For this reason, and in line with some previous studies searching for COPD markers [52,69], we decided to choose asymptomatic smokers without airflow limitation for the control group, normalizing our results by recent exposure to tobacco (objectivated through the blood levels of cotinine, an unbeatable marker of the presence and intensity of the smoking habit) [75]. We have to recognize, however, that some of these factors considered as “confounding” in our study design may interact with the respiratory disease, causing metabolic changes of mixed origin. A new study with a different design would be required to elucidate on which changes this would occur. Our results, therefore, leave that question open.

### 3.4. Final Metabolomic Signature for COPD (Excluding Xenobiotics)

The combination of only 10 differentially abundant metabolites between patients and controls has allowed an approximation to a reasonable suspicion of having COPD. This metabolomic signature may potentially be used in the near future to facilitate blood-based screening of possible COPD cases in either the general or at-risk populations, especially considering the high prevalence and underdiagnosis of this respiratory disease [4,5,6,7,8,9,10,11,12]. The use of a screening panel could help guide the application of spirometry in high-risk populations, extending its use beyond traditional criteria such as heavy smoking history or findings from questionnaires or computed tomography (CT) scans, as has occasionally been proposed [76,77,78,79]. It should be noted that spirometry is a functional test requiring specific equipment, a well-trained technician and full collaboration from the subject to ensure technical quality and diagnostic reliability. These requirements make its large-scale implementation difficult and likely contribute to underdiagnosis. In contrast, a biological risk signature, such as the one proposed in the present study, could eventually lead to a simple and widely applicable test. However, further validation in independent cohorts is necessary before clinical implementation can be considered.

The “COPD signature” included several fatty acids and their conjugates. Some LCFAs showed reduced levels in patients, whereas one MCFAs were more abundant when compared with HC. Although partly speculative, the low levels of the former might reflect a reduced capacity for energy production via beta-oxidation [80,81], while the higher levels of the latter, deriving from leucine metabolism, could be interpreted as a sign of enhanced catabolic activity [82]. Similarly, the elevated levels of N-methylglutamate, a derivative of glutamic acid, may point to changes in amino acid catabolism in COPD. Conversely, lower levels of urocanate could reflect insufficient histidine availability in such a population, which might influence allergic or vasodilatory responses [83]. These interpretations remain hypothetical and require further investigation.

In addition to these findings, two metabolites in the signature were linked to carbohydrate metabolism, and specifically to the pentose phosphate pathway: glyceric acid, which was decreased in COPD patients, and gluconic acid, with was elevated compared to controls. High levels of gluconic acid may suggest an up-regulation of initial glucose catabolism [84], while lower levels of glyceric acid could indicate a possible bottleneck in later glycolytic steps [85]. Again, this imbalance might result in decreased energy production, along with impaired functions of glyceric acid, such as stimulation of protein synthesis and facilitation of oxygen release by hemoglobin. In the context of impaired pulmonary gas exchange and the frequent presence of anemia in COPD patients, this could hypothetically contribute to altered aerobic metabolism.

As for DEA, which was found at relatively high levels in COPD patients, it competes with ethanolamine in the formation of glycerophospholipids [86]. These lipids are critical structural elements of cell membranes, involved in anchoring proteins, signaling and the synthesis of eicosanoids, as well as being a part of the pulmonary surfactant. Elevated DEA levels could potentially lead to the generation of abnormal glycerophospholipids, such as phosphatidyl-DEA instead of phosphatidylethanolamine, which may display altered metabolic behavior. Finally, reduced levels of 2-aminonicotinic acid, an amide of nicotinic acid included in the vitamin B3 complex, were also observed in COPD patients [87]. This reduction might be linked to decreased synthesis of steroid hormones, NAD and NADPH, possibly contributing to redox imbalance, impaired formation of lipids and nucleic acids, and once again, reduced energy generation.

### 3.5. Strengths and Potential Limitations

One of the greatest strengths of the present study is that it included a relatively large series of patients and matched controls, whose biological samples were carefully obtained and preserved in reference research centers and hospitals belonging to the Spanish Network of Excellence for Research in Respiratory Diseases (CIBERES). In addition, controls and patients shared a history of smoking, a relevant factor that has not always been considered in previous studies. Moreover, the influence of active smoking was assessed and our metabolic data normalized based on a very objective element: the blood concentration of cotinine. The adjustment made for this and other potential confounding factors (sex, age, nutritional status), which could potentially have affected our initial results, is another of the strengths of the present study. Finally, the use of two complementary (semi-targeted and untargeted) metabolomic approaches for the identification of metabolites present in the blood, is an additional strength facilitating discovery of new potential biomarkers.

One of the most obvious potential limitations of our study is the impossibility of identifying the origin and potential metabolic functions of xenobiotics. However, this is probably a minor issue since a robust and novel metabolomic signature could be obtained after the exclusion of these markers. Another limitation is that the links between our findings and their potential metabolic consequences are presented as interpretative hypotheses, and therefore will require validation through future targeted mechanistic and longitudinal studies.

## 4. Materials and Methods

### 4.1. Participants and Ethics

This is a case-control study that included 91 clinically stable COPD patients, and 91 controls (HC, asymptomatic smokers without airflow limitation) obtained from two multicenter cohorts recruited from 11 teaching hospitals and primary care centers in Spain from January 2015 to May 2018 (the BIOMEPOC and the EARLY COPD projects) [88,89]. Both studies were approved by the local ethics committees (refs. 2014/5695/l and 2014/5895/I, respectively) and the investigation was conducted in accordance with the Declaration of Helsinki and its recent updates. Patients were randomly chosen from both cohorts, and HCs were subsequently selected to ensure a similar distribution of age, sex and nutritional status. All participants were Caucasians and signed a written informed consent prior to any clinical data or sample collection. The diagnosis of COPD was based on a history of tobacco smoking and/or other harmful exposures as well as the presence of airflow obstruction after bronchodilation (FEV_1_/FVC < 0.7) [90], whereas HCs were present or former smokers, asymptomatic and with normal spirometry. All patients were clinically stable (absence of exacerbations for at least 3 months prior to entering the study). Exclusion criteria included treatments with systemic corticosteroids and the presence of other chronic lung diseases. Further details on all clinical procedures performed on the two cohorts have been published elsewhere [88,89]. The present manuscript has followed the STROBE guidelines.

### 4.2. Collection of Blood Samples

After fasting overnight, whole blood was collected by peripheral venipuncture in ethylenediaminetetraacetic acid (EDTA) tubes and centrifuged at 1200× *g* for 15 min within 1 h after extraction to obtain plasma. Samples were then aliquoted and stored at −80 °C until the metabolomic analysis.

### 4.3. Metabolomics Procedure

#### 4.3.1. Metabolite Extraction

Plasma samples were initially thawed on ice and vortexed. A QC-pool was prepared by aliquoting 10 µL together from each sample; and 10 µL more from each one were diluted in 190 µL of a precooled (−20 °C) in methanol:acetonitrile 75:25 extraction solution. The latter was prepared with isotope-labeled internal standards 5 µM glucose 13C6 (ref. CLM-1396-5), 1 µM glutamine 13C5 (CLM-1822-H-PK), 0.5 µM pyruvate 13C3554 (CLM-2440-1), 2.5 µM glutamate 13C5 (CLM-3949-0.25), 2.5 µM alanine 13C1 (CLM-116-1) and 2.5 µM lactate 13C3 (CLM-1579-0.5) [all from Cambridge Isotope Laboratories, Tewksbury, MA, USA]. The samples were vortexed for 10 min at 4 °C to extract metabolites and centrifuged for another 15 min at maximal speed to pellet proteins and any particulate matter. Then, 80 µL of the supernatant was transferred into high-performance liquid chromatography (HPLC) glass vials with glass inserts and stored at −80 °C until the LC-MS analysis. The extraction procedure was additionally controlled for contaminants by procedure blanks of extraction buffer. The QC-pool samples were injected every 10 individual samples and used to evaluate Intra Run variations.

#### 4.3.2. LC-MS Run Parameters

This analysis was conducted as previously described [91], with only minor adaptations. Briefly, the Thermo Vanquish Flex ultra-high-performance liquid chromatography (UPLC) system coupled to Orbitrap Exploris 240 Mass Spectrometer (both from Thermo Fisher Scientific, Waltham, MA, USA) were used, having a resolution of 120,000 at 200 mass/charge ratio (*m*/*z*), electrospray ionization and polarity switching mode to enable both positive and negative ions across a mass range of 67–1000 *m*/*z*. The UPLC setup consisted in ZIC-pHILIC column (SeQuant; 150 mm × 2.1 mm, 5 µm; Merck, Rahway, NJ, USA). Then, 5 µL of plasma extracts were injected, and the compounds were separated on a mobile phase gradient for 15 min, starting with a combination of 20% aqueous (20 mM ammonium carbonate adjusted to 9.2 pH, with 0.1% of 25% ammonium hydroxide) and 80% organic (acetonitrile) compounds, and terminating with 20% acetonitrile. Flow rate and column temperature were maintained at 0.2 mL/min and 45 °C, respectively, for a total run time of 27 min. Thermo Xcalibur 4.4 (also from Thermo Fisher Scientific) was used for data acquisition.

#### 4.3.3. Metabolite Identification and Quantification

##### Semi-Targeted

The identification of different metabolites was first obtained using a semi-targeted approach. Shortly, peak areas of each metabolite were determined using the Thermo TraceFinderTM 5.1 software (also from Thermo Fisher Scientific). Metabolites were identified by the exact mass of the singly charged ion and by their known retention time, using an in-house multi-scale (MS) library (around 600 metabolites) built by running commercial standards of all detected metabolites.

##### Untargeted

To expand the metabolome coverage, an additional untargeted analysis was also conducted using the Compound Discoverer software version 3.3 (also from Thermo Fisher Scientific). In this case, the alignment of retention times across all data files was performed via the ChromeAlign node, employing a pooled sample (consisting of an amalgamate of aliquots from all biological specimens, subjected to repeated injections and occurring every 10 samples) as a reference file for quality control and normalization of batch effects. The detection of unknown compounds (with a minimum peak intensity threshold of 1 × 105 AUC), and the subsequent grouping of compounds was performed across all samples with the following primary parameters (all other settings were maintained at their default values): a mass tolerance of 5 ppm, a retention time tolerance of 0.2 min for compound detection, focusing solely on M + H and M − H ions, and a peak rating filter configured to 4. Missing values were addressed by using the software’s fill gap feature (mass tolerance of 5 ppm and a signal-to-noise tolerance of 1.5). The “Search Mass Lists” node was incorporated for identification, leveraging an in-house metabolite library that encompasses retention times.

The annotation of metabolites was conducted in a hierarchical manner, reflecting diminishing confidence levels: (1) by correlating the mass and retention time of the observed signal with those found in an in-house library that was constructed using commercial standards (mass tolerance of 5 ppm and a retention time tolerance of 0.5 min); (2) by matching fragmentation spectra against the advanced mass spectral database mzCloud (www.mzcloud.org, accessed on 7–14 May 2024), with precursor and fragment mass tolerances set to 10 ppm and a match factor threshold of 80; and (3) for compounds devoid of fragmentation spectra data; annotations were performed utilizing the Human Metabolome DataBase (HMDB, www.hmdb.ca), BioCyc Genome Database Collection (www.biocyc.org) and the Kyoto Encyclopedia of Genes and Genomes (KEGG, www.genome.jp/kegg) (all accessed on 7–14 May 2024), with the application of filtering criteria that required the mzlogic score of the software to exceed 50 and to possess fewer than three potential candidates [92]. Finally, the identified candidates were subjected to a manual review to eliminate false identification and enhance the quality of data.

With these two complementary approaches 461 metabolites were identified, but only 360 were present in at least 80% of plasma samples and were included in the final analysis.

### 4.4. Data Analysis

#### 4.4.1. Statistical Analysis of Clinical Data

Data are presented as median (Interquartile range, IQR) or mean ± standard deviation (SD) for continuous variables, with absolute and relative frequencies for categorical ones. Quantitative variables were first tested for normality using the Kolmogorov–Smirnov test before applying any other test. Comparison of general and clinical data between COPD patients and HC were analyzed by either t-tests or, when appropriate, the non-parametric Mann–Whitney U test. All these analyses were performed using the Statistical Package for the Social Sciences (version 25.0) (SPSS, IBM. Chicago, IL, USA).

#### 4.4.2. Metabolomic Analysis

To normalize and obtain total measurable ion peak intensities for each sample, raw data files were processed with Compound Discoverer 3.3 (also from Thermo Fisher Scientific). Each identified metabolite intensity was normalized to the total intensity of the sample, and the raw data were then filtered to only include compounds with <20% missingness over samples. Filtered features were subsequently uploaded to MetaboAnalyst 6.0 (University of Alberta, AB, Canada) [93] and missing values were imputed using 1/5 of the minimum positive value. All raw data were log2-transformed to approximate a normal distribution and scaled using Pareto’s algorithm.

All metabolomic data were evaluated using both univariate and multivariate statistical approaches using the above-mentioned MetaboAnalyst 6.0 software. Univariate analysis was first carried out to obtain an overview of potentially altered metabolites in COPD patients relative to HC, and establish those features included in a supervised multivariate approach. Identification of DAMs between COPD patients and controls was carried out using two-sided unpaired *t*-tests, and false discovery rate (FDR) correction by the Benjamini–Hochberg method was performed on p values to account for multiple comparisons. Metabolites with FDR < 0.1 were considered as significantly different between the two groups, and were used for subsequent steps. The volcano plot was used to visualize metabolite differences between both groups. Next, considering COPD as the principal variable, multiple linear regression was adopted to analyze DAMs between groups, also accounting for factors previously known to potentially influence the metabolism [i.e., age, sex, nutrition (represented by BMI) and recent smoking status (based on the blood levels of cotinine, a metabolite of nicotine that quantifies recent cigarette smoke exposure)] [17,18,20,21,47,94]. The Small Molecule Pathway Database (SMPDB, www.smpdb.ca), KEGG and HMDB (all accessed on 1–30 November 2024) were used to identify the most relevant biochemical pathways, providing the initial framework for metabolomic data. Since one of the main purposes of the study was to identify potentially human metabolites potentially useful for screening, xenobiotics (chemical compounds considered as not naturally produced by human beings) were excluded for the final analysis.

Furthermore, to investigate the most significant DAMs metabolites in plasma, a supervised machine learning analysis was performed by SVM for the predictive model. The ROC curves were obtained to verify which metabolite signature had the highest sensitivity and specificity for COPD.

## 5. Conclusions

In summary, the present study clearly demonstrates that there is a metabolic signature characteristic of COPD patients. This metabolic signature is composed of fatty acids, amino acid and carbohydrate metabolites, a pseudoglycerolipid and vitamin B3, suggesting that COPD patients experience alterations in energy production, the redox balance systems and synthesis of various key molecules belonging to relevant metabolic pathways. Therefore, it suggests new and/or complementary pathophysiological mechanisms involved in the disease, with potential implications for future therapies. Moreover, given the current challenges related to the underdiagnosis of COPD, our results suggest that a panel of just ten metabolites could be used for screening purposes, identifying population at risk. A subsequent forced spirometry would confirm or exclude the presence of the disease.

## Figures and Tables

**Figure 1 ijms-26-04526-f001:**
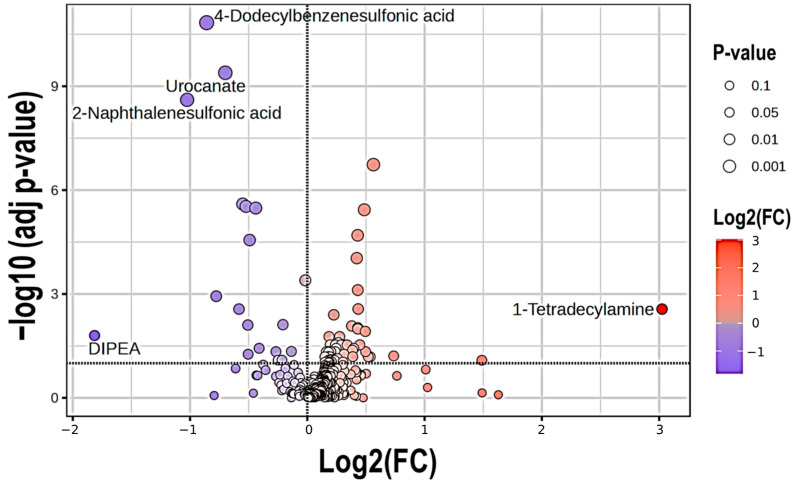
Volcano plot of the most over- and under-represented metabolites present in COPD patients as compared to HC. The dashed horizontal line indicates the limit of significance. Red circles denote over-represented metabolites (right), whereas blue circles indicate under-represented metabolites (left, which are separated by the dashed vertical line.

**Figure 2 ijms-26-04526-f002:**
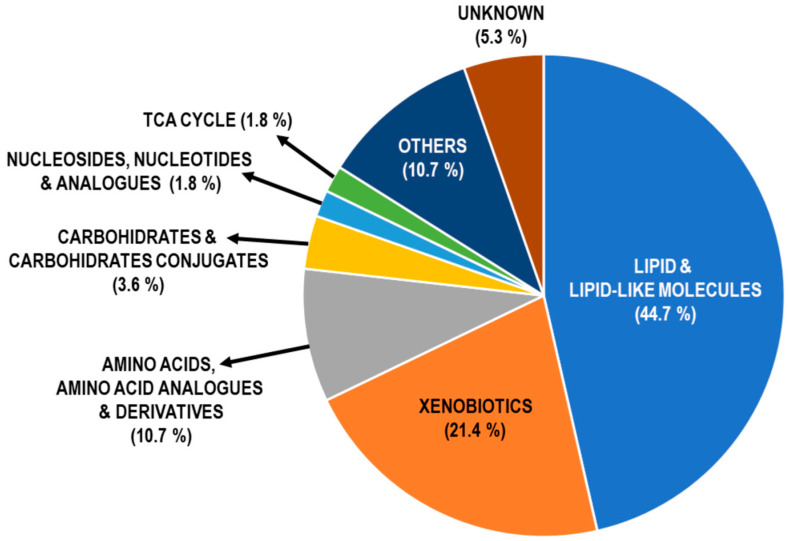
Distribution profile of differential metabolites upon adjusting for age, sex, BMI and recent smoking status. Abbreviation: TCA, tricarboxylic acid cycle.

**Figure 3 ijms-26-04526-f003:**
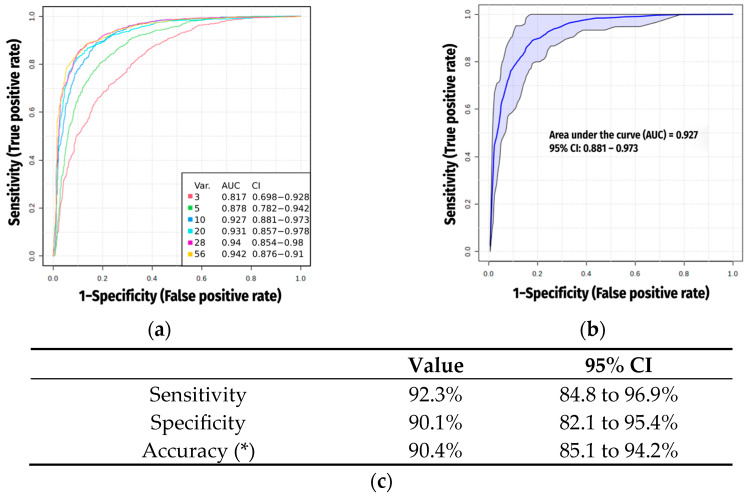
(**a**) Receiver operating characteristic (ROC) curves from all models generated by Support Vector Machine (SVM), showing the area under the curve (AUC) values; (**b**) ROC curve from the best model (which included 10 metabolites). The blue line represents the average curve across 100 Monte Carlo cross-validation (MCCV) iterations, while the shaded area illustrates the variability in model performance across MCCV runs; (**c**) Comprehensive evaluation metrics. (*) Accuracy was calculated based on a COPD prevalence of 11.8% in the Spanish adult population [12].

**Figure 4 ijms-26-04526-f004:**
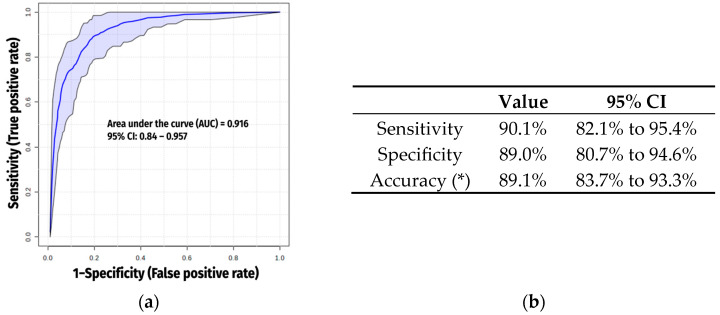
(**a**) Receiver operating characteristic (ROC) curve generated by the SVM model, with the area under the curve (AUC) calculated from DAMs when excluding xenobiotics. The blue line represents the average curve across 100 Monte Carlo cross-validation (MCCV) iterations, while the shaded area illustrates the variability in model performance across MCCV runs; (**b**) Comprehensive evaluation metrics. (*) Accuracy was calculated based on a COPD prevalence of 11.8% in the Spanish adult population [12].

**Table 1 ijms-26-04526-t001:** Baseline characteristics of subjects included in the analysis.

	HC (N = 91)	COPD (N = 91)
Age, median (IQR), yr.	48 (44–61)	50 (46–66)
Male, N (%)	50 (55)	46 (51)
Body Mass Index (kg/m^2^), mean ± SD	26.5 ± 4.4	26.1 ± 5.5
Current or former smokers, N (%)	91 (100)	91 (100)
Pack Years, median (IQR)	23 (15–32)	39 (24–60) ***
Post-BD FEV_1_ (% pred.), median (IQR)	95 (85–104)	56 (43–76) ***
FEV_1_/FVC ratio, median (IQR)	82 (74–87)	56 (40–65) ***
DLco, median (IQR)	85 (79–96)	56 (43–76) ***
GOLD class, N (%)		
GOLD 1	-	19 (21)
GOLD 2	-	37 (41)
GOLD 3	-	24 (26)
GOLD 4	-	11 (12)

Values are expressed as median (interquartile range, IQR) or mean ± SD for continuous variables, and as N (percentage) for categorical variables. Significances: ***, *p* < 0.001 compared to HC. Abbreviations: HC, controls (asymptomatic smokers without airflow limitation); COPD, Chronic Obstructive Pulmonary Disease; Post-BD FEV_1_, post-bronchodilator Forced Expiratory Volume in one second; FVC, Forced Vital Capacity; DLco, Diffusion capacity for carbon monoxide; GOLD, Global Initiative for Chronic Obstructive Lung Disease.

**Table 2 ijms-26-04526-t002:** Panel of the best 10 metabolites discriminating COPD from controls.

Compound Name	logFC	HMDB	Chemical Taxonomy (Super Class)	Chemical Taxonomy (Sub Class)	KEGG Pathways
Hexadecanoic acid (Palmitic acid)	−0.057507	HMDB0000220	Lipids and lipid-like molecules	Fatty acids and conjugates (LCFA)	Fatty acid Metabolism
Glyceric acid	−0.32747	HMDB0000139	Organic Oxygen Compounds	Carbohydrates and their conjugates	Pentose phosphate pathway
Urocanate	−0.45756	HMDB0000301	Organoheterocyclic Compounds	Imidazoles	Histidine metabolism
2-aminonicotinic acid	−0.32725	HMDB0061680	Organoheterocyclic Compounds	Pyridinecarboxylic acids and derivatives	
2-Hydroxyisocaproic acid (Leucic acid)	0.33133	HMDB0000665	Lipids and Lipid-like Molecules	Fatty acids and conjugates (MC Hydroxy fatty acid)	Fatty acid Metabolism
Diethanolamine	0.16912	HMDB0004437	Organic Nitrogen Compounds	Amines	Glycerophospholipid metabolism
1-Tetradecylamine	3.0246	HMDB0258887	Organic Nitrogen Compounds	Amines	
Pentapropylene glycol (PPG n5)	0.5634		Organic Oxygen Compounds	Alcohols and polyols	
2-Naphthalenesulfonic acid	−1.0249	HMDB0255446	Benzenoids	Naphthalene sulfonic acids and derivatives	
4-Dodecylbenzenesulfonic acid	−0.85833	HMDB0059915	Benzenoids	Benzene sulfonic acids and derivatives	

Abbreviations: FC, Fold Change; HMDB, Human Metabolome Database; KEEG, Kyoto Encyclopedia of Genes and Genomes; LCFA, long-chain fatty acid; MC, medium-chain (fatty acid).

**Table 3 ijms-26-04526-t003:** Panel of the best 10 metabolites used for discrimination between COPD and controls excluding xenobiotics.

Compound Name	logFC	HMDB	Chemical Taxonomy (Super Class)	Chemical Taxonomy (Sub Class)	KEGG Pathways
Hexadecanoic acid (Palmitic acid)	−0.057507	HMDB0000220	Lipids and Lipid-like Molecules	Fatty acids and conjugates (LCFA)	Fatty acid Metabolism
Glyceric acid	−0.32747	HMDB0000139	Organic Oxygen Compounds	Carbohydrates and Carbohydrate conjugates	Pentose phosphate pathway
Urocanate	−0.45756	HMDB0000301	Organoheterocyclic Compounds	Imidazoles	Histidine metabolism
2-Aminonicotinic acid	−0.32725	HMDB0061680	Organoheterocyclic Compounds	Pyridinecarboxylic acids and derivatives	
2-Hydroxyisocaproic acid (Leucic acid)	0.33133	HMDB0000665	Lipids and Lipid-like Molecules	Fatty acids and conjugates (MC Hydroxy fatty acid)	Fatty acid Metabolism
Diethanolamine	0.16912	HMDB0004437	Organic Nitrogen Compounds	Amines	Glycerophospholipid Metabolism
Gluconic acid	0.24119	HMDB0000625	Organic Oxygen Compounds	Carbohydrates and their conjugates	Pentose phosphate pathway
2-Hydroxytetradecanoic acid (2-Hydroxymyristic acid)	−0.20222	HMDB0002261	Lipids and Lipid-like Molecules	Fatty acids and conjugates (LC Hydroxy fatty acid)	Fatty acid Metabolism
14-Methylhexadecanoic acid	−0.12759	HMDB0031067	Lipids and Lipid-like Molecules	Fatty acids and conjugates (LC chain Methyl fatty acid)	Fatty acid Metabolism
N-Methylglutamate	0.14171	HMDB0062660	Organic Acids and Derivatives	Amino acids, peptides and analogues	Derivative of Glutamic acid

Abbreviations: FC, Fold Change; HMDB, Human Metabolome Database; KEEG, Kyoto Encyclopedia of Genes and Genomes; LCFA, long-chain fatty acid; MC, medium-chain (fatty acid).

## Data Availability

The original contributions presented in this study are included in the article/Appendix A. Further inquiries can be directed to the corresponding author.

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
