# Peer review of "Metabolomic Plasma Profile of Chronic Obstructive Pulmonary Disease Patients"

_ijms, 2025, doi:10.3390/ijms26104526_

Round 1
Reviewer 1 Report
Comments and Suggestions for Authors
This manuscript presents a well-executed metabolomic study identifying a distinct plasma metabolite signature capable of discriminating patients with chronic obstructive pulmonary disease (COPD) from matched healthy smokers. The authors employ both semi-targeted and untargeted LC-MS/MS-based metabolomics, followed by robust statistical modeling and machine learning algorithms to derive biologically meaningful insights and propose a potential screening biomarker panel for COPD. The study is of high relevance given the high global prevalence of COPD and its underdiagnosis.
- While the authors extensively discuss xenobiotics, including plausible environmental, microbial, and pharmaceutical sources, the interpretation remains largely speculative. Although a xenobiotic-free model was constructed, the presence of these compounds, particularly industrial pollutants and surfactants, demands cautious interpretation and perhaps additional validation.
- Although the authors make a compelling effort to link metabolic perturbations to pathophysiological processes (e.g., energy inefficiency, inflammation, protein degradation), much of the mechanistic interpretation remains speculative. Tone down pathophysiological assertions or clearly state them as hypotheses. Suggest validation in mechanistic or longitudinal studies.
- The proposed 10-metabolite panel shows excellent discriminative power (AUC = 0.916), but no external or cross-validation cohort is included. The robustness of the panel in independent populations remains unknown.
- Maybe the authors could replace informal terms like “healthy smokers” with more precise language (e.g., “asymptomatic smokers without airflow limitation”).
Author Response
To the Reviewer 1.
This manuscript presents a well-executed metabolomic study identifying a distinct plasma metabolite signature capable of discriminating patients with chronic obstructive pulmonary disease (COPD) from matched healthy smokers. The authors employ both semi-targeted and untargeted LC-MS/MS-based metabolomics, followed by robust statistical modeling and machine learning algorithms to derive biologically meaningful insights and propose a potential screening biomarker panel for COPD. The study is of high relevance given the high global prevalence of COPD and its underdiagnosis.
We thank the reviewer for his kind words on our manuscript.
- While the authors extensively discuss xenobiotics, including plausible environmental, microbial, and pharmaceutical sources, the interpretation remains largely speculative. Although a xenobiotic-free model was constructed, the presence of these compounds, particularly industrial pollutants and surfactants, demands cautious interpretation and perhaps additional validation.
We fully agree with the reviewer. We have been especially careful in interpreting the findings related to xenobiotics in the new version of the manuscript, also mentioning the necessity of further validation studies (see new section 1.2.2).
- Although the authors make a compelling effort to link metabolic perturbations to pathophysiological processes (e.g., energy inefficiency, inflammation, protein degradation), much of the mechanistic interpretation remains speculative. Tone down pathophysiological assertions or clearly state them as hypotheses. Suggest validation in mechanistic or longitudinal studies.
We have modified our wording in accordance with the reviewer's recommendations (see mostly the Abstract and Discussion sections in the new version of the manuscript). Thus, we have avoided being overly assertive and/or clearly stating that these are mere hypotheses whenever we attempt to link our findings to pathophysiological processes involved in COPD. Again, we also mention that mechanistic or longitudinal studies, which are needed to corroborate our hypotheses.
- The proposed 10-metabolite panel shows excellent discriminative power (AUC = 0.916), but no external or cross-validation cohort is included. The robustness of the panel in independent populations remains unknown.
Totally agreed, and this is mentioned repeatedly now throughout the revised version of the manuscript.
- Maybe the authors could replace informal terms like “healthy smokers” with more precise language (e.g., “asymptomatic smokers without airflow limitation”).
Totally agreed. We have implemented this change throughout the revised manuscript.
Reviewer 2 Report
Comments and Suggestions for Authors
Submitted manuscript nicely executed and presented work. However my specific comments are bellow:
- Many metabolomic study of COPD available in literature (https://pmc.ncbi.nlm.nih.gov/articles/PMC10613990/), Author should justify the relevance of this metabolomics of COPD study.
- Author should add a heat map of all altered metabolites in COPD patients.
- 2. Author should add a bar graph to show changes in metabolites which has presented in table 2 and table 3.
- Xenobiotics is one of the important contributing factor in disease etiology. Author may add study of xenobiotics in COPD patients and control blood sample.
Author Response
Point-by-Point responses to the Reviewers’ comments
Reviewer 2.
Submitted manuscript nicely executed and presented work. However my specific comments are bellow:
- Many metabolomic study of COPD available in literature (https://pmc.ncbi.nlm.nih.gov/articles/PMC10613990/), Author should justify the relevance of this metabolomics of COPD study.
As suggested by the reviewer, the new and distinctive aspects of our study compared to previous literature have been emphasized in the revised version of the manuscript.
2. Author should add a heat map of all altered metabolites in COPD patients.
A heat map has been added in a Supplementary File of the new version of the manuscript. Although we understand that it may help complement the information from a visual perspective, it is partly redundant with the data already presented in the text and tables. Therefore, we consider more appropriate to include it as supplementary material.
3. Author should add a bar graph to show changes in metabolites which has presented in table 2 and table 3.
We have added these figures requested by the reviewer to the supplementary material, as the revised version of the manuscript already contains this information and has a substantial number of figures.
4. Xenobiotics is one of the important contributing factor in disease etiology. Author may add study of xenobiotics in COPD patients and control blood sample.
This part of the Discussion has been fully rewritten in accordance with the reviewer's suggestion, while aiming to avoid excessive hypothesis or speculation, given that the available information on these compounds and their potential relevance to the disease -particularly in the case of possible environmental contaminants or surfactants- is limited or controversial. Please also refer to our response to Reviewer 1 on his/her similar comment.
